# ON THE SEQUENCE EVALUATION BASED ON STOCHASTIC PROCESSES

## ABSTRACT

Generative models have gained significant prominence in Natural Language Processing (NLP), especially in tackling the complex task of modeling and evaluating long text sequences. This task is crucial for advancing various downstream applications, such as text generation and machine translation. Recent methods that utilize stochastic processes to capture the intrinsic dynamics of sequences have shown superior performance in generative modeling. However, the accurate encoding of both temporal and structural dependencies from text datasets, as well as leveraging this encoded information for sequence evaluation, remains an open area of research. In this paper, we propose a novel approach to learn the stochastic dynamics of long text sequences, utilizing a negative log-likelihood-based encoder that outperforms contrastive learning methods. We also introduce a likelihood-based evaluation metric for long-text assessment, which measures sequence coherence and can be applied to downstream tasks such as Human-AI discrimination. Our encoder preserves sequence coherence effectively and performs robustly on out-of-domain datasets. Additionally, the proposed evaluation metric captures both temporal and structural information comprehensively. Theoretical analysis demonstrates the superiority of our metric in sequence evaluation, and experimental results highlight its flexibility and exceptional performance across a variety of tasks, showcasing its utility in diverse NLP applications.

## 1 INTRODUCTION

Generative models are rapidly gaining traction in NLP (Zou et al., 2023; Yang et al., 2023; Yi et al., 2024), particularly in tackling the intricate task of modeling and generating long text sequences. This challenge is critical for various downstream applications, including text generation and machine translation. Recently, the use of stochastic representations to model latent spaces has emerged as a promising solution, demonstrating considerable potential in diverse fields such as time-series analysis (Liu et al., 2021), dynamical flow modeling (Albergo et al., 2023; Albergo & Vanden-Eijnden, 2023), and video generation (Zhang et al., 2023). In the domain of text generative models, Wang et al. (2022) proposed a method that frames long sequences as stochastic dynamical flows, achieving competitive results in generating coherent long texts. However, the challenge of accurately learning the time-dependent probability density functions embedded within text datasets remains unresolved. Moreover, fully harnessing the information encoded in the stochastic representation continues to pose a significant challenge that has yet to be comprehensively addressed.

**Brownian Bridge for stochastic representation modeling and coherence metric.** Linguistic theory offers valuable insights into modeling long articles as stochastic processes. For example, coherent long texts tend to exhibit certain common properties (Sheng et al., 2024): (1) a central theme that remains consistent throughout the text, (2) sentences at the beginning and end that emphasize this central idea, and (3) sentences in the middle that may diverge slightly from the theme but still remain aligned with it. These characteristics naturally align with the concept of a "bridge" in stochastic processes, where the structure supports a consistent trajectory while allowing for controlled variability. In this work, we employ the Brownian Bridge (abbreviated as BB) to represent this conceptual bridge. It is a special stochastic process characterized by "fixed" starting and ending positions (Øksendal & Øksendal, 2003) and has been widely applied in various fields. Inspired by the work of Sheng et al. (2024) and Wang et al. (2022), we discover that the fit of a trajectory as a

BB can be used to evaluate and represent several characteristics of long-form text as an *evaluation metric* and *text encoder*.

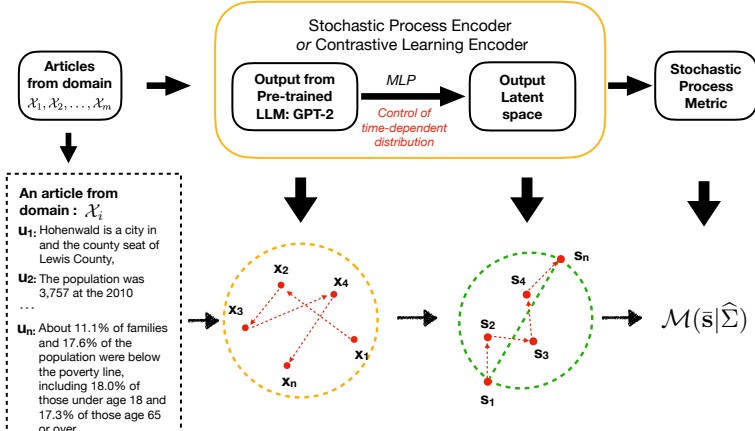

Figure 1: Schematic diagram of the Stochastic Representation and Metric. An article from domain $\mathcal{X}_i$, segmented into sentences $(u_1, u_2, \cdots u_n)$, is processed by the Stochastic Process (or Contrastive Learning) encoder, which consists of a pre-trained large language model (LLM) and a multi-layer perceptron (MLP). The LLM generates a raw representation $(x_1, x_2, \cdots x_n)$, which the MLP transforms into a time-dependent latent trajectory $(s_1, s_2, \cdots s_n)$. Additionally, we introduce a stochastic process metric (SPM) that captures the temporal and structural properties of this latent trajectory.

First, we introduce a novel likelihood-based evaluation metric, Stochastic Process Metric (SPM) for long-text assessment that quantifies coherence by leveraging both temporal and structural dependencies of the stochastic process, with detailed explanation in Section 2.1. Unlike similar approaches that rely heavily on the training domain (Lai & Tetreault, 2018; Jeon & Strube, 2022) and are limited by their training schemes – which typically evaluate only pairwise data and restrict comparisons to articles of the same length, SPM offers greater flexibility while maintaining comparable performance. To demonstrate this, we generalize the established Shuffle test (Barzilay & Lapata, 2005; Joty et al., 2018; Moon et al., 2019) into a Mixed Shuffle test. Instead of merely comparing shuffled and unshuffled versions of the same article, this new test also compares across the shuffle and unshuffled versions between different articles, which can be used to evaluate the robustness of the metric regardless of individual article properties, e.g. article length. Moreover, SPM proves useful in downstream tasks such as Human-AI discrimination and performs robustly in out-of-domain (O.O.D.) tasks, likely due to it capture the structural information of the stochastic representation.

Second, we present a novel text encoder, Stochastic Process Encoder (SP Encoder) for effectively capturing the stochastic dynamics present in extensive collections of long articles. We propose a negative log-likelihood-based training objective for text encoders, comparing its effectiveness to that of contrastive learning methods (Wang et al., 2022). The SP Encoder translates sequences into a generalized BB while retaining essential information, leading to improved performance on O.O.D. tests. In contrast, while contrastive learning can be useful for assessing coherence, it may fall short in adequately capturing the representation of the target stochastic process, which could hinder its effectiveness on other tasks, particularly those involving O.O.D. data.

In summary, the main contributions of our work are as follows:

- We propose a novel likelihood-based metric (SPM), designed to capture both temporal and structural dependencies with a solid theoretical foundation. The SPM demonstrates superior flexibility and effectiveness in sequence evaluation, as evidenced by its performance across various tasks.

- We hypothesized and validated that fitting the stochastic representation, as measured by the SPM, can serve as an effective and flexible metric for multiple downstream tasks, such as distinguishing between human and AI-generated writing.

- We introduced a negative log-likelihood-based method (SP Encoder), for learning stochastic representations that preserves sequence coherence while efficiently utilizing latent dimensions. Compared to contrastive learning encoders, it exhibits robust and superior performance across O.O.D. datasets.

## 2 METHOD

### 2.1 STOCHASTIC REPRESENTATION

In this section, we introduce a stochastic representation of the encoded sequences by modeling them using BBs, which are commonly utilized stochastic processes in various applications. We begin by defining a standard BB $\{B(t) : t \in [0, T]\}$ with $B(0) = 0$ and $B(T) = 0$. For any $t \in [0, T]$, the process $B(t)$ follows a normal distribution $B(t) \sim N(0, t(T - t)/T)$. Additionally, for $s, t \in [0, T]$ with $s < t$, the covariance between $B(s)$ and $B(t)$ is given by $\text{Cov}(B(s), B(t)) = s(T - t)/T$. A general BB can then be constructed as $a + (t/T)(b - a) + \sigma B(t)$, where $a$ and $b$ are fixed start and end points, respectively, and $\sigma$ is the standard deviation of the process.

For an input sequence $\bar{\mathbf{s}} = (s_0, \ldots, s_T)$ with $s_t \in \mathbb{R}^d$ for $t = 0, 1, \ldots, T$, we capture temporal dependence using standard BBs. To account for structural dependence among the components, we consider $d$ independent standard BBs $B_1(t), \ldots, B_d(t)$ over the interval $[0, T]$. At each time $t$, the sequence is modeled as $s_t = \mu_t + \mathbf{W}(B_1(t), \ldots, B_d(t))^\top$, where $\mathbf{W} \in \mathbb{R}^{d \times d}$ is a transformation matrix and $\mu_t = s_0 + (t/T)(s_T - s_0)$ represents the mean at time $t$. The structural dependence is captured by $\Sigma = \mathbf{W}\mathbf{W}^\top$. Let $\mathbf{s} = (s_1, \ldots, s_{T-1})$ denote the sequence excluding the start and end points, and let $\boldsymbol{\mu} = (\mu_1, \ldots, \mu_{T-1})$ be the corresponding means.

The SP Encoder and the SPM are based on the likelihood function of the input sequences, with $\Sigma$ being the only unknown parameter. The following proposition presents the likelihood function.

**Proposition 1.** *Let $\Sigma_T \in \mathbb{R}^{(T-1) \times (T-1)}$ be the covariance matrix with entries $[\Sigma_T]_{s,t} = s(T - t)/T$. For $n$ independent input sequences $\bar{\mathbf{s}}_1, \ldots, \bar{\mathbf{s}}_n$ with lengths $T_1 + 1, \ldots, T_n + 1$, generated by the same $\mathbf{W}$ (or equivalently, $\Sigma$), the log-likelihood function is*

$$\ell(\Sigma | \{\bar{\mathbf{s}}_i\}_{i=1}^n) = \frac{d \sum_{i=1}^n (T_i - 1)}{2} \log(2\pi) - \frac{d}{2} \sum_{i=1}^n \log(|\Sigma_{T_i}|) - \frac{\sum_{i=1}^n (T_i - 1)}{2} \log(|\Sigma|)$$

$$- \frac{1}{2} \sum_{i=1}^n \text{tr}(\Sigma^{-1}(\mathbf{s}_i - \boldsymbol{\mu}_i)\Sigma_{T_i}^{-1}(\mathbf{s}_i - \boldsymbol{\mu}_i)^\top).$$

By Proposition 1, we derive the log-likelihood function given the input sequences, which enables us to compute the maximum likelihood estimate (MLE) of $\Sigma$.

**Proposition 2.** *Under the setting of Proposition 1, the MLE of $\Sigma$ given $\{\bar{\mathbf{s}}_i\}_{i=1}^n$ is*

$$\widehat{\Sigma} = \Big( \sum_{i=1}^n (T_i - 1) \Big)^{-1} \Big( \sum_{i=1}^n (\mathbf{s}_i - \boldsymbol{\mu}_i)\Sigma_{T_i}^{-1}(\mathbf{s}_i - \boldsymbol{\mu}_i)^\top \Big).$$

The training of the SP Encoder and the definition of the SPM are thus based on the MLE of $\Sigma$.

### 2.2 ENCODERS

The encoder architecture consists of two components: a frozen, pretrained GPT-2 model from the Hugging Face library and a trainable multi-layer perceptron (MLP) network. We extract the hidden state corresponding to the end-of-sentence (EOS) token from the last layer of the GPT-2 model. This hidden state serves as the input to a four-layer MLP, which is trained to map the input into the latent space. The purpose of the encoder is to learn a nonlinear mapping from the raw input space to the latent space, denoted as $f_\theta : \mathcal{X} \to \mathcal{S}$. In practice, we use two loss functions to train the encoder: the contrastive learning loss $L_{\text{CL}}$ and the negative log-likelihood loss $L_{\text{NLL}}$. The contrastive learning loss $L_{\text{CL}}$ is primarily designed to enhance the encoder's ability to distinguish between positive and negative samples from the data, following the approach in (van den Oord et al., 2018). In contrast, the negative log-likelihood loss $L_{\text{NLL}}$ directly enforces the latent space representations to follow

a specified distribution according to the likelihood function. While Wang et al. (2022) utilized contrastive learning for the encoder, our SP Encoder employs the negative log-likelihood loss, which offers certain advantages over their approach, both theoretically and empirically. For comparison purposes, we first introduce the contrastive learning encoder as presented in Wang et al. (2022).

### 2.2.1 Contrastive Learning Encoder (CL Encoder)

In the context of Wang et al. (2022), a structure assumption $\Sigma = \mathbf{I}_d$ was imposed, where $\mathbf{I}_d$ is the $d$-dimensional identity matrix. For arbitrary starting point $\mathbf{s}_0$ at time $t = 0$ and ending point $\mathbf{s}_T$ at time $t = T$, the marginal distribution of $\mathbf{s}_t$ at time $t$ is given by $\mathbf{s}_t \mid \mathbf{s}_0, \mathbf{s}_T \sim N((1 - t/T)\mathbf{s}_0 + (t/T)\mathbf{s}_T, [t(T - t)/T]\mathbf{I}_d)$. Consider any triplet of observations $(\mathbf{x}_1, \mathbf{x}_2, \mathbf{x}_3)$ with $\mathbf{x}_1, \mathbf{x}_2, \mathbf{x}_3 \in \mathcal{X}$. The goal is to ensure that $f_\theta(\mathbf{x}_2)$ follows the above marginal distribution with starting point $f_\theta(\mathbf{x}_1)$ and ending point $f_\theta(\mathbf{x}_3)$. For a sequence of observations $(\mathbf{x}_0, \ldots, \mathbf{x}_T)$, let $B = \{(\mathbf{x}_0, \mathbf{x}_t, \mathbf{x}_T)\}$ be a batch consisting of randomly sampled positive triplets $(\mathbf{x}_0, \mathbf{x}_t, \mathbf{x}_T)$ with $0 < t < T$. Then, the contrastive learning loss function is defined as

$$L_{\mathrm{CL}} = \mathbb{E}\left[ -\log \frac{\exp(d(\mathbf{x}_0, \mathbf{x}_t, \mathbf{x}_T; f_\theta))}{\sum_{(\mathbf{x}_0, \mathbf{x}_{t'}, \mathbf{x}_T) \in B} \exp(d(\mathbf{x}_0, \mathbf{x}_{t'}, \mathbf{x}_T; f_\theta))} \right],$$

where

$$d(\mathbf{x}_0, \mathbf{x}_t, \mathbf{x}_T; f_\theta) = -(2\sigma^2)^{-1}\|f_\theta(\mathbf{x}_t) - (1 - t/T)f_\theta(\mathbf{x}_0) - (t/T)f_\theta(\mathbf{x}_T)\|_2^2, \ \ \sigma^2 = t(T - t)/T.$$

However, there are two theoretical drawbacks to this contrastive learning approach. First, the assumption $\Sigma = \mathbf{I}_d$ implies independence and homogeneity among the dimensions of the encoded sequence, which may not capture the true structural dependencies present in the sequence. Second, using the marginal distribution neglects the time-dependent structure of the stochastic process, limiting the model's ability to capture temporal dependencies.

Empirically, it has been observed that only one dimension of the encoder's output is effective, which is undesirable for representing complex sequences. To address these issues and incorporate both structural and temporal dependencies, we propose our novel SP Encoder, which minimizes the negative log-likelihood loss. This approach allows us to model the latent space more accurately by considering the full covariance structure and the temporal dynamics of the data, leading to improved theoretical foundations and empirical performance over the contrastive learning method.

### 2.2.2 Stochastic Process Encoder (SP Encoder)

Consider a multi-domain problem with $m$ domains $\mathcal{X}_1, \mathcal{X}_2, \ldots, \mathcal{X}_m$ each associated with domain-specific true structural parameters $\Sigma_1, \Sigma_2, \ldots, \Sigma_m$, respectively. For each domain $\mathcal{X}_j$, we have $n_j$ independent raw inputs $\mathbf{x}_{j1}, \ldots, \mathbf{x}_{jn_j}$. We define the encoded sequences as $\bar{\mathbf{s}}_{ji}^\theta = f_\theta(\mathbf{x}_{ji})$ for $j = 1, \ldots, m$ and $i = 1, \ldots, n_j$, where $f_\theta$ is the encoder parameterized by $\theta$. When the encoder parameters reach their optimal values $\theta^*$, the sequences $[\mathbf{s}_{ji}^{\theta^*}]_{i=1}^{n_j}$ are expected to be i.i.d. samples from BBs with parameters $\Sigma_j$ for each domain $\mathcal{X}_j$.

We employ the negative log-likelihood (NLL) as the loss function to train the encoder. According to Proposition 1, for each $\theta$, the negative log-likelihood for domain $\mathcal{X}_j$ depends on $\Sigma_j$ and the inputs $[\mathbf{x}_{ji}]_{i=1}^{n_j}$ through the expression $\sum_{i=1}^{n_j}(T_i - 1)\log(|\Sigma_j|) + \sum_{i=1}^{n_j} \mathrm{tr}(\Sigma_j^{-1}(\mathbf{s}_i^\theta - \boldsymbol{\mu}_i^\theta)\Sigma_{T_i}^{-1}(\mathbf{s}_i^\theta - \boldsymbol{\mu}_i^\theta)^\top)$. We consider the following training process.

**Batch Processing:** We divide the inputs $[\mathbf{x}_{ji}]_{i=1}^{n_j}$ into several batches. For each batch $\mathcal{B}$, we compute the batch loss using the current estimate $\widehat{\Sigma}_j$ of $\Sigma_j$: $\sum_{i \in \mathcal{B}} \mathrm{tr}(\widehat{\Sigma}_j^{-1}(\mathbf{s}_i^\theta - \boldsymbol{\mu}_i^\theta)\Sigma_{T_i}^{-1}(\mathbf{s}_i^\theta - \boldsymbol{\mu}_i^\theta)^\top)$. This loss function measures how well the encoded sequences fit the assumed BB model with the current structural parameter estimate.

**Handling Large Sequences:** When the sequence lengths $T_i$ are large, computing the full loss can be computationally intensive. To address this, we randomly sample a triplet of time points $t = (t_1, t_2, t_3)$ with $1 \le t_1 < t_2 < t_3 \le T_i - 1$. We extract the corresponding sub-matrices $[\mathbf{s}_i^\theta]_t$ and $[\boldsymbol{\mu}_i^\theta]_t$ of size $d \times 3$ from $\mathbf{s}_i^\theta$ and $\boldsymbol{\mu}_i^\theta$, respectively. Let $[\Sigma_{T_i}]_t$ be the $3 \times 3$ sub-matrix of $\Sigma_{T_i}$ corresponding to the selected time points. The loss for each $i$ in the batch becomes $\mathrm{tr}(\widehat{\Sigma}_j^{-1}([\mathbf{s}_i^\theta]_t - [\boldsymbol{\mu}_i^\theta]_t)[\Sigma_{T_i}]_t^{-1}([\mathbf{s}_i^\theta]_t - [\boldsymbol{\mu}_i^\theta]_t)^\top)$. This approach reduces computational complexity while still capturing temporal dependencies at selected time points.

**Updating Structural Parameters:** After processing all batches for $\mathcal{X}_j$, we update the estimate of $\widehat{\Sigma}_j$ using the MLE: $\widehat{\Sigma}_j = [\sum_{i=1}^{n_j}(T_i-1)]^{-1}[\sum_{i=1}^{n_j}(\mathbf{s}_i^\theta - \boldsymbol{\mu}_i^\theta)\Sigma_{T_i}^{-1}(\mathbf{s}_i^\theta - \boldsymbol{\mu}_i^\theta)^\top]$. This update aggregates information from all sequences in the domain to refine the structural parameter estimate.

**Regularization for Stability:** To stabilize the training process, we regularize $\widehat{\Sigma}_j$ by blending it with a scaled identity matrix. We compute the average variance $\widehat{\sigma}_j^2$ and update $\widehat{\Sigma}_j$ as follows, using a small regularization parameter $\epsilon > 0$: $\widehat{\Sigma}_j = (1-\epsilon)[\sum_{i=1}^{n_j}(T_i-1)]^{-1}[\sum_{i=1}^{n_j}(\mathbf{s}_i^\theta - \boldsymbol{\mu}_i^\theta)\Sigma_{T_i}^{-1}(\mathbf{s}_i^\theta - \boldsymbol{\mu}_i^\theta)^\top] + \epsilon\widehat{\sigma}_j^2\mathbf{I}_d$ with $\widehat{\sigma}_j^2 = [\sum_{i=1}^{n_j}(T_i-1)d]^{-1}[\sum_{i=1}^{n_j}\mathrm{tr}((\mathbf{s}_i^\theta - \boldsymbol{\mu}_i^\theta)\Sigma_{T_i}^{-1}(\mathbf{s}_i^\theta - \boldsymbol{\mu}_i^\theta)^\top)]$. This regularization shifts $\widehat{\Sigma}_j$ slightly towards isotropy, improving numerical stability during optimization.

**Total Empirical Loss Function:** After iterating over all domains, the total empirical loss function becomes

$$L_{\mathrm{NLL}} = \sum_{j=1}^{m}\sum_{i=1}^{n_j}(T_i-1)\log(|\Sigma_j|) + \sum_{j=1}^{m}\sum_{i=1}^{n_j}\mathrm{tr}(\Sigma_j^{-1}(\mathbf{s}_i^\theta - \boldsymbol{\mu}_i^\theta)\Sigma_{T_i}^{-1}(\mathbf{s}_i^\theta - \boldsymbol{\mu}_i^\theta)^\top).$$

Minimizing this loss over $\theta$ encourages the encoder to produce sequences that align with the assumed stochastic process model across all domains.

Compared to the CL Encoder, our SP Encoder offers three key improvements. First, by directly minimizing the negative log-likelihood, we ensure that the encoded sequences adhere to the BB model. This principled approach leverages the full statistical structure of the sequence, leading to more accurate modeling of the latent space. Second, unlike CL Encoder that relies on marginal distributions, our approach considers the joint distribution of entire sequences. This allows us to capture temporal dependencies inherent in the stochastic process, improving the representation of the sequence. Third, we avoid assuming independence and homogeneity among latent dimensions. By estimating the covariance matrix, the encoder learns the structural dependencies between dimensions, enabling it to represent complex stochastic dynamics effectively, especially in long sequences.

## 2.3 STOCHASTIC PROCESS METRIC (SPM)

Consider the sequence $\bar{\mathbf{s}} = (s_0, \ldots, s_T)$, with $\mathbf{s}$ and $\boldsymbol{\mu}$ defined as before. To evaluate the coherence of the sequence from a domain with unknown parameters $\Sigma$, a natural approach is to compute its density under the assumed model. If $\bar{\mathbf{s}}$ is a BB with covariance $\Sigma$, then by Proposition 1, the log density of $\bar{\mathbf{s}}$ is given by:

$$\log p(\bar{\mathbf{s}}|\Sigma) = -\frac{d(T-1)}{2}\log(2\pi) - \frac{d}{2}\log(|\Sigma_T|) - \frac{(T-1)}{2}\log(|\Sigma|) - \frac{1}{2}\mathrm{tr}(\Sigma^{-1}(\mathbf{s}-\boldsymbol{\mu})\Sigma_T^{-1}(\mathbf{s}-\boldsymbol{\mu})^\top).$$

If the sequence is not coherent with the model, the density would be low because the time-dependent structure is violated. However, the log density is not scale-invariant with respect to the sequence length. This means that for two sequences of different lengths generated from the same domain, their log densities are not directly comparable. Additionally, the log density can take any real value, whereas a positive and standardized metric is desirable for practical purposes. To address these issues, we note that $\mathrm{vec}(\mathbf{s}-\boldsymbol{\mu}) \sim N(0, \Sigma_T \otimes \Sigma)$, where $\mathrm{vec}(\cdot)$ denotes the vectorization operator and $\otimes$ denotes the Kronecker product. Then we have $[\Sigma_T \otimes \Sigma]^{-1/2}\mathrm{vec}(\mathbf{s}-\boldsymbol{\mu}) \sim N(0, \mathbf{I}_{(T-1)\times d})$, which implies $\|[\Sigma_T \otimes \Sigma]^{-1/2}\mathrm{vec}(\mathbf{s}-\boldsymbol{\mu})\|^2 = \mathrm{vec}(\mathbf{s}-\boldsymbol{\mu})^\top[\Sigma_T \otimes \Sigma]^{-1}\mathrm{vec}(\mathbf{s}-\boldsymbol{\mu}) \sim \chi_{(T-1)\times d}^2$, where $\chi_{(T-1)d}^2$ denotes the chi-square distribution with $(T-1)d$ degrees of freedom. Since $\mathrm{vec}(\mathbf{s}-\boldsymbol{\mu})^\top[\Sigma_T \otimes \Sigma]^{-1}\mathrm{vec}(\mathbf{s}-\boldsymbol{\mu}) = \mathrm{tr}(\Sigma^{-1}(\mathbf{s}-\boldsymbol{\mu})\Sigma_T^{-1}(\mathbf{s}-\boldsymbol{\mu})^\top)$, we can define our metric, the SPM, by normalizing with the degrees of freedom.

**Definition** (SPM). *Let $\widehat{\Sigma}$ be the estimate of $\Sigma$ from Proposition 2. The Stochastic Process Metric (SPM) is defined as*

$$\mathcal{M}(\bar{\mathbf{s}}|\widehat{\Sigma}) = \mathrm{tr}(\widehat{\Sigma}^{-1}(\mathbf{s}-\boldsymbol{\mu})\Sigma_T^{-1}(\mathbf{s}-\boldsymbol{\mu})^\top)/[(T-1)d].$$

The SPM is always positive. When $\widehat{\Sigma}$ accurately estimates $\Sigma$ and $\bar{\mathbf{s}}$ is a BB with covariance $\Sigma$, the SPM will concentrate around 1. A larger value of SPM indicates that the sequence is less likely to have been generated by the given covariance matrix $\Sigma$.

Our metric, SPM, is novel in two key aspects. First, by utilizing the temporal covariance matrix $\Sigma_T$, the SPM captures the time-dependent structure inherent in the sequence, which is essential for accurately assessing sequence coherence. Second, the inclusion of the covariance matrix $\Sigma$ allows the SPM to account for structural dependencies among the latent dimensions, providing a more comprehensive evaluation of the sequence's adherence to the assumed stochastic process.

## 3 EXPERIMENTS AND PROBLEMS

We now design experiments to evaluate the performance of SPM and the effectiveness of the SP Encoder in learning stochastic representations. Specifically, we aim to address the following three key questions (Q):

- **Q1**: *Can SPM evaluate both local and global coherence?* In Section 4.1, we demonstrate that SPM not only excels in basic shuffle and unshuffle tasks but also achieves outstanding results when comparing articles of varying lengths—an evaluation that current state-of-the-art methods cannot perform effectively.

- **Q2**: *Can SPM handle the human-AI discrimination task?* In Section 4.2, we show that SPM can effectively distinguish between human-written text and text generated by large language models. Compared to recent well-known methods, SPM delivers competitive results with shorter inference times and a more flexible computation scheme.

- **Q3**: *Can the SP Encoder enhance the learning of stochastic representations?* In Section 4.3, we demonstrate that the SP Encoder outperforms the CL Encoder in learning stochastic representations by effectively using latent dimensions and their correlations. Moreover, this advantage leads to more robust performance when applied to unseen datasets.

### 3.1 DATASETS

**WikiSection:** We use dataset introduced in Arnold et al. (2019) which contains selected Wikipedia articles on the topic of global cities and have clear topic structures. Each article in this collection follows a pattern certain sections such as abstract, history, geographics and demographics. The training split contains 2165 articles and the test split has 658 articles.

**HC3:** The Human ChatGPT Comparison Corpus (HC3) (Guo et al., 2023) includes comparative responses from human experts and ChatGPT, covering questions from various fields such as open-domain, finance, medicine, law, and psychology.

**WikiText:** WikiText language modeling dataset (Merity et al., 2016) is a much larger set of verified good and featured articles extracted from Wikipedia compared to WikiSection, and in section F in the appendix, we further compare these two dataset and show that there is only $\sim 1\%$ potential overlap in topics. We used *WikiText-103-v1* collection in specific for experiments. This dataset encompass over 100 million tokens from 29,061 full articles. The dataset is assessible through Huggingface [1].

**WSJ:** The Wall Street Journal (WSJ) corpus (Consortium, 1994) is part of the Penn Treebank dataset and contains 24 sections, each comprising 100 news articles. These articles were selected from daily publications spanning 1987 to 1989. Following previous work (Moon et al., 2019), sections 00–13 were used as the training set, while sections 14–24 were used as the test set. After filtering out articles with fewer than 10 sentences, the training set includes 716 articles, and the test set consists of 573 articles.

### 3.2 TASKS

We assess the effectiveness of our proposed methods across both artificial and real-world downstream tasks. In artificial scenarios, we introduce incoherence into human-authored documents to evaluate the ability of SPM to detect these inconsistencies and accurately rank the original and shuffled documents based on coherence across various experimental configurations. Additionally, we investigate the applicability of this method in a realistic context to determine whether the learned temporal patterns can effectively differentiate between human-generated and AI-generated content.

---

[1]`https://huggingface.co/datasets/EleutherAI/WikiText_document_level`

Furthermore, we compare the model's learning performance under two objective functions (CL and SP) across both in-domain and out-of-domain settings. The following sections detail the setups used in these experiments.

**Global and Local discrimination** For global discrimination tasks, we applied the Shuffle test (Barzilay & Lapata, 2005; Moon et al., 2019), which involves randomly permuting sentences within a discourse to create an incoherent document, which is then compared to the original version. Specifically, we shuffled the entire article with varying sentence block sizes (1, 2, 5, and 10). For local discrimination tasks, following Moon et al. (2019), we randomly sampled a number of 3-sentence windows, denoted as $w$, and sentences within each window were shuffled to create perturbed copies. For these two tasks, 20 shuffled copies were generated for each article, with duplicates discarded.

**Mixed Shuffled test** Building on the Shuffle test, we introduced a more complex variant where we compare the score of an unshuffled article with shuffled articles from the entire dataset, rather than just its own shuffled version. A robust and general-purpose coherence score should consistently identify the unshuffled article as more coherent across these comparisons.

**Human-AI text discrimination** In the Human-AI discrimination tasks, we trained both the CL Encoder on the **WikiSection** dataset, then applied them to generate latent representations for articles from HC3. To differentiate between human and AI content, we computed the SPM for both human and AI-generated articles, hypothesizing that, since WikiSection is based on human-written text, human-authored articles in HC3 should have a lower SPM than AI-generated ones. Based on this hypothesis, we conducted the discrimination task.

**Out-of-domain (O.O.D.) test** In this task, we trained the SP and CL Encoder on the WikiSection and evaluated their performance on the Shuffle Task with data generated from WikiSection and WSJ.

## 4 RESULTS

### 4.1 COHERENCE EVALUATION TASKS

As shown in Tables 1, we first implement global and local discrimination tasks on WikiSection. In the global tasks, SPM significantly outperforms the BBScore and matches SOTA results. (See Appendix C for more details on methods we compared to.) The SOTA method, developed using a complex network structure and trained on unshuffle-shuffle data pairs, serves as a robust baseline. To achieve comparable results, we adopted a similar approach to that in Sheng et al. (2024), incorporating a multi-layer perceptron to fully leverage SPM across various shifting windows, which is denoted by "SPM + CLS". Our results demonstrate that SPM surpasses the SOTA method in global discrimination tasks with larger block sizes, underscoring its potential to capture more globalized properties. However, despite outperforming the BBScore, our score falls short in local discrimination tasks. This suggests that SPM is more effective in capturing global characteristics rather than responding to local perturbations. Given that our metric is based on a likelihood approximation of the entire trajectory, it is reasonable that it struggles to capture finer, localized features.

| Methods | Accuracy (Global) | | | | Accuracy (Local) | | |
|---|---|---|---|---|---|---|---|
| | $\mathcal{D}_{b=1}$ | $\mathcal{D}_{b=2}$ | $\mathcal{D}_{b=5}$ | $\mathcal{D}_{b=10}$ | $\mathcal{D}_{w=1}$ | $\mathcal{D}_{w=2}$ | $\mathcal{D}_{w=3}$ |
| ENTITY GRID (Barzilay & Lapata, 2005) | 85.73 | 82.79 | 75.81 | 64.65 | 53.04 | 60.83 | 66.67 |
| UNIFIED COHERENCE (Moon et al., 2019) | **99.73** | **97.86** | 96.90 | 96.09 | **77.47** | **82.98** | **87.87** |
| BBSCORE (Sheng et al., 2024) | 83.39 | 80.71 | 79.36 | 78.66 | 50.29 | 60.15 | 64.06 |
| SPM (OURS) | 95.06 | 94.72 | 95.13 | 95.67 | 56.35 | 68.83 | 74.24 |
| SPM + CLS (OURS) | 97.55 | 97.17 | **97.42** | **97.36** | 73.72 | 73.30 | 77.76 |

Table 1: Results of Global and Local Discrimination tasks on WikiSection. $\mathcal{D}_{b=i}$, $i = 1, 2, 5, 10$ refers to datasets constructed with varying levels of block shuffling. $\mathcal{D}_{w=j}$, $j = 1, 2, 3$ denotes datasets with different number of windows in local perturbation.

In global and local shuffle tasks, most current high-performance methods, including the SOTA approach, rely on pairwise training and are unable to effectively compare articles of different lengths, as these models are typically constructed based on sentence-wise matching and comparisons. However, in the Mixed Shuffle test which evaluate the metric robustness across different articles, as

shown in Table 9, SPM surpasses these SOTA method by generating a metric that can be compared across different articles. We use the basic entity-grid method (Barzilay & Lapata, 2005) as a baseline and the result highlights that our score enables article-wise comparison. It also demonstrates significant potential in more complex tasks. Additionally, SPM outperforms the BBScore in this article-wise comparison, underscoring a key contribution of our design—mitigating the effect of article length on score evaluation. This property allows for a more general and robust comparison across diverse articles. By visualizing the score distribution across different article lengths (as shown in Fig 4 in the appendix), we can clearly observe this effect.

| Methods | Mixed Shuffled Test | | | |
|---|---|---|---|---|
| | $\mathcal{D}_{b=1}$ | $\mathcal{D}_{b=2}$ | $\mathcal{D}_{b=5}$ | $\mathcal{D}_{b=10}$ |
| ENTITY GRID (Barzilay & Lapata, 2005) | 46.10 | 52.29 | 53.69 | 63.02 |
| BBSCORE (Sheng et al., 2024) | 22.37 | 24.94 | 23.84 | 19.69 |
| SPM (OURS) | **90.32** | **86.03** | **79.26** | **77.89** |

Table 2: Accuracy of Mixed Shuffled Test tasks on WikiSection.

## 4.2 HUMAN-AI DISCRIMINATION TASKS

In this task, we hypothesize that human writing, compared to AI-generated text, displays temporal dynamics and structural patterns similar to those observed in other human-written articles. Specifically, we propose that an encoder trained on a human-written dataset will more accurately capture the characteristics of human writing than those of AI-generated text, resulting in a lower likelihood and smaller SPM for human-authored content. To test this hypothesis, we use the dataset from Guo et al. (2023), as summarized in Table 3. The data from Guo et al. (2023), which includes both human and AI writing, is treated as unseen data for the encoder trained on WikiSection.

| Methods | HC3 (without Q&A) | HC3 (with Q&A) |
|---|---|---|
| BBSCORE (Sheng et al., 2024) | 37.53 | 31.47 |
| DETECTGPT (Mitchell et al., 2023) | 64.30 | 63.30 |
| SPM (OURS) | **70.67** | **69.71** |

Table 3: Accuracy of the Human-AI discrimination task.

Our results indicate that SPM performs well on both the Human-AI (without Q&A) and Human-AI (Q&A) benchmarks, demonstrating its effectiveness. This strength is particularly notable as SPM is training-free and relies solely on comparing article dynamics and writing structure. To further explore the importance of capturing correlations in a high-dimensional latent space, we also evaluated BBScore's performance in the Human-AI discrimination task. As shown in Table 3, SPM consistently outperforms BBScore, which we attribute to BBScore's limited ability to capture the complex structural information essential for understanding the latent space.

Additionally, to highlight the flexibility and competitive performance of SPM compared to LLM-based models, we assessed the perturbation discrepancy metric proposed in DetectGPT (Mitchell et al., 2023), a method known for its high performance in AI detection tasks. Our results reveal that SPM surpasses DetectGPT when using a comparable number of model inferences. DetectGPT's performance is influenced by a hyperparameter—the number of perturbations—which directly affects both the number of model inferences and the computational complexity. As shown in Table 8 in the Appendix, we tested cases with 1 and 10 perturbations. With 1 perturbation, DetectGPT's accuracy was approximately 64%, lower than SPM's 70%, while requiring twice the number of model inferences per text. With 10 perturbations, DetectGPT's accuracy increased to 84%, but this required 11 model inferences per text, making it significantly more computationally intensive than SPM.

## 4.3 COMPARISON BETWEEN SP ENCODER AND CL ENCODER

We first compare the performances of SP and CL Encoder in learning stochastic representations and examine how these differences impact downstream tasks. By visualizing the covariance matrix ($\widehat{\Sigma}$) of the SP and CL Encoder in Figures 2 (A) and (B), we find $\widehat{\Sigma}$ of CL Encoder exhibits only one

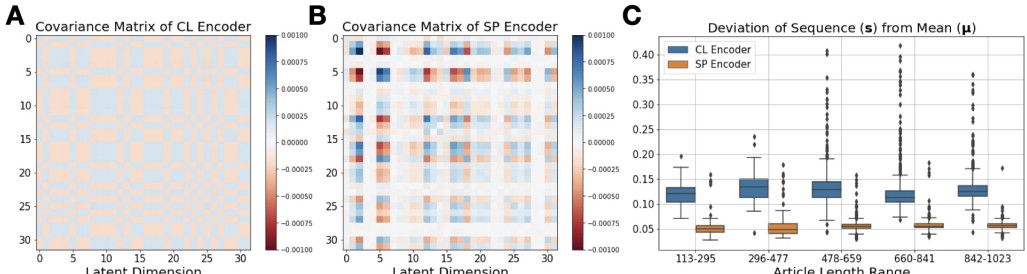

Figure 2: CL vs SP Encoder in the latent representation. (A) and (B) depict the approximated covariance matrix ($\widehat{\Sigma}$) from the latent representation of CL and SP Encoder, respectively. (C) shows the deviation of $\mathbf{s}$ from $\boldsymbol{\mu}$, and it implies the robustness of the representation generated by SP Encoder.

effective dimension, while $\widehat{\Sigma}$ of SP Encoder use more latent dimensions with heterogeneity (see Fig 5 (A.2) and (B.2) in the appendix for further evidence on ineffective dimensions in the CL Encoder, with a specific latent representation). As discussed in Section 2.2, learning the covariance structure is crucial for learning the complex stochastic representations, and poor covariance modeling can result in suboptimal downstream performance. To explore these differences further, we evaluate two tasks: 1) Shuffle test and Mixed Shuffle test on **WikiSection** with SPM, and 2) O.O.D performance analysis on datasets (**WikiText** and **WSJ**) with encoder trained on **WikiSection**.

| Tasks | Shuffle Test tasks | | | | Mixed Shuffle Test tasks | | | |
|---|---|---|---|---|---|---|---|---|
| | $\mathcal{D}_{b=1}$ | $\mathcal{D}_{b=2}$ | $\mathcal{D}_{b=5}$ | $\mathcal{D}_{b=10}$ | $\mathcal{D}_{b=1}$ | $\mathcal{D}_{b=2}$ | $\mathcal{D}_{b=5}$ | $\mathcal{D}_{b=10}$ |
| CL ENCODER | 95.06 | 94.72 | 95.13 | 95.67 | 90.32 | 86.03 | 79.26 | 77.89 |
| SP ENCODER | 94.42 | 92.90 | 90.77 | 86.69 | 78.60 | 67.14 | 55.19 | 53.97 |

Table 4: Comparison of CL and SP Encoders on In-Domain tasks. Results show the performance (accuracy) of the SPM computed with latent trajectories encoded by the CL and SP Encoders. Encoders were trained on the WikiSection training set and evaluated on the WikiSection test set.

As shown in Table 4, the CL encoder excels in both Shuffle test tasks and Mixed Shuffle test tasks, effectively distinguishing between shuffled and unshuffled articles as well as different articles and their shuffled counterparts. However, as shown in Table 5, the SP Encoder shows better performance in O.O.D. tasks. In the O.O.D. task, we evaluate SPM on an O.O.D. dataset by training two encoder (CL and BB Encoder) on WikiSection, seperately, and then applying it to compute the latent trajectory of articles from unseen datasets (WikiText and WSJ). Next, we assess performance on the Shuffle test task, and the results show that SPM achieve high performance in O.O.D. tests, with the SP Encoder outperforming the CL encoder. It indicates that the SP Encoder effectively learns both the distribution and the dynamics of the entire article, and it can help SPM to robustly evaluate and identify O.O.D. stochastic representations. To further support this O.O.D. result, we compare it with BBScore (Sheng et al., 2024) and unified coherence (Moon et al., 2019).

| Methods | | Shuffle Test tasks (WikiText) | | | | Shuffle Test tasks (WSJ) | | | |
|---|---|---|---|---|---|---|---|---|---|
| Score type | Encoder | $\mathcal{D}_{b=1}$ | $\mathcal{D}_{b=2}$ | $\mathcal{D}_{b=5}$ | $\mathcal{D}_{b=10}$ | $\mathcal{D}_{b=1}$ | $\mathcal{D}_{b=2}$ | $\mathcal{D}_{b=5}$ | $\mathcal{D}_{b=10}$ |
| UNIFIED COHERENCE (Moon et al., 2019) | —— | 60.02 | 9.63 | 44.80 | 66.51 | 56.29 | 53.55 | **57.89** | **61.46** |
| BBSCORE (Sheng et al., 2024) | CL Encoder | 70.32 | 72.09 | 76.84 | 77.73 | 36.61 | 38.90 | 41.53 | 49.24 |
| BBSCORE (Sheng et al., 2024) | SP Encoder | 55.97 | 58.07 | 51.13 | 52.46 | 45.16 | 43.26 | 36.91 | 35.36 |
| SPM (OURS) | CL Encoder | 91.30 | 87.22 | **86.14** | 88.18 | 52.69 | 53.86 | 50.32 | 51.11 |
| SPM (OURS) | SP Encoder | **93.03** | **89.32** | 83.02 | **88.31** | **59.96** | **68.24** | 50.78 | 50.00 |

Table 5: Results of the O.O.D. Task. Encoders were trained on the WikiSection and evaluated on Shuffle Test tasks using the WikiText and WSJ to assess their performance on O.O.D. tasks.

## 5 RELATED WORK

Stochastic processes have demonstrated robust capabilities in modeling complex tasks across various fields, including biology (Horne et al., 2007) and finance (Øksendal & Øksendal, 2003). Recently,

in the realm of machine learning, the use of stochastic representations to model latent spaces has shown considerable promise in diverse applications such as time-series analysis (Liu et al., 2021) and dynamical flow modeling (Albergo et al., 2023; Albergo & Vanden-Eijnden, 2023). Notably, such methods excel in generation tasks, including video generation (Zhang et al., 2023), and long text generation (Wang et al., 2022). A critical aspect of these tasks is the accurate learning of stochastic representations from datasets, which requires capturing the time-dependent probability density functions embedded within real-world data. Generally, there are two approaches to tackle this challenge. One method is the likelihood-free training paradigm (Durkan et al., 2020), exemplified by contrastive learning techniques, which have demonstrated significant effectiveness in handling high-dimensional data (van den Oord et al., 2018). This approach enables the learning of predictive density indirectly, rather than through direct reconstruction (Mathieu et al., 2021). The alternative method is the traditional likelihood-based approach, such as stochastic interpolants (Albergo et al., 2023; Albergo & Vanden-Eijnden, 2023), which requires the pre-definition of specific target stochastic processes. Both methods exhibit substantial potential in their respective tasks.

Coherence, as described by Reinhart (1980), defines the discourse structure of a text, where high-quality writing should exhibit a logical flow with well-connected ideas and topics. Previous studies have indicated that neural language models, particularly transformers, often struggle to effectively capture coherence structures (Deng et al., 2022). To better guide language models in learning the dynamics of long texts, methods that construct latent spaces have been developed (Bowman et al., 2016; Gao et al., 2021). These methods aim to model sentence embeddings by leveraging information from neighboring utterances. However, most of these approaches yield static representations and overlook the dynamic nature of the entire text. A recent study introduced the concept of using stochastic representations, specifically the Brownian bridge, to incorporate "temporal dynamics" into modeling latent space (Wang et al., 2022). This approach demonstrates superior performance in modeling long-range text dependencies and achieving notable results in generating long, coherent texts from learned latent plans. However, their training scheme may lead to ineffective dimensions and fail to fully capture the structural and temporal information encoded in the dataset.

In addition to latent space modeling, automatically assessing coherence in any given text remains a challenge (Sheng et al., 2024; Maimon & Tsarfaty, 2023). Building on stochastic concepts, Sheng et al. (2024) developed a heuristic metric for coherence assessment, grounded in the unsupervised learning approach proposed by Wang et al. (2022). This score demonstrated considerable performance on artificial shuffle tasks. However, their method relies on a heuristic understanding of the Brownian bridge and fails to adequately establish a theoretical foundation for their metric setup, which limit the effectiveness and flexibility of their score, particularly its sensitivity to article length.

## 6 CONCLUSION

In this paper, we investigate the structural and temporal properties encoded in the stochastic representation of latent trajectories and their applications in NLP tasks, both theoretically and empirically. First, we propose a novel, flexible coherence evaluation metric, SPM, that is not affected by individual article properties, such as text length. To validate this, we design a Mixed Shuffle test based on the established Shuffle test. We also discover that our metric, which assesses the fit of target stochastic processes, can aid in distinguishing between human-written and AI-generated data, indicating that the stochastic representation encodes properties useful for human-AI discrimination. Second, we introduce a negative log-likelihood-based training method (SP Encoder) to enhance the learning of stochastic representations. Our results show that the SP Encoder outperforms in O.O.D. tasks, due to its effectiveness in using the latent dimensions as suggested by the covariance matrix.

For future work, we aim to conduct a thorough analysis of how to leverage SPM for multi-domain tasks, such as domain identification, and to utilize its text-length-independent property to develop text generation models that maintain semantic meaning while being robust across varying text lengths. Additionally, compared to existing well-known Human-AI scores, our method is simpler and more flexible, achieving comparable performance with similar inference times; however, it may become inferior as their model complexity increases, such as much more inference times. Thus, extending this method is an intriguing avenue for exploration. Furthermore, a deeper theoretical understanding of the differences between likelihood-based and contrastive learning-based models is another direction for future research.

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

# A Proofs

## A.1 Proof of Proposition 1

*Proof.* We fix the start and end points $s_0$ and $s_T$ and calculate the likelihood function of the input sequence $\mathbf{s}$.

Given that $s_t - \mu_t = \mathbf{W}(B_1(t), \ldots, B_d(t))^\top$, and considering the independence of $B_1(t), \ldots, B_d(t)$ along with the properties of the standard BB, we have for any $t, t' \in \{1, 2, \ldots, T-1\}$: $\mathbb{E}[s_t - \mu_t] = 0$, $\mathrm{Var}[s_t] = [\Sigma_T]_{t,t}\Sigma$ and $\mathrm{Cov}[s_t, s_{t'}] = [\Sigma_T]_{t,t'}\Sigma$. Therefore, the vectorized form of $\mathbf{s} - \boldsymbol{\mu}$ follows a multivariate normal distribution:

$$\mathrm{vec}(\mathbf{s} - \boldsymbol{\mu}) \sim N(0, \Sigma_T \otimes \Sigma),$$

where $\mathrm{vec}(\cdot)$ denotes vectorization and $\otimes$ represents the Kronecker product.

Using the likelihood function of the multivariate normal distribution, we have:

$$L(\Sigma|\bar{\mathbf{s}}) = (2\pi)^{-d(T-1)/2}|\Sigma_T \otimes \Sigma|^{-1/2}\exp[-\mathrm{vec}(\mathbf{s} - \boldsymbol{\mu})^\top[\Sigma_T \otimes \Sigma]^{-1}\mathrm{vec}(\mathbf{s} - \boldsymbol{\mu})/2].$$

Using properties of the Kronecker product, we have $|\Sigma_T \otimes \Sigma| = |\Sigma_T|^d|\Sigma|^{T-1}$ and then

$$\mathrm{vec}(\mathbf{s} - \boldsymbol{\mu})^\top[\Sigma_T \otimes \Sigma]^{-1}\mathrm{vec}(\mathbf{s} - \boldsymbol{\mu}) = \mathrm{vec}(\mathbf{s} - \boldsymbol{\mu})^\top[\Sigma_T^{-1} \otimes \Sigma^{-1}]\mathrm{vec}(\mathbf{s} - \boldsymbol{\mu})$$

$$= \mathrm{vec}(\mathbf{s} - \boldsymbol{\mu})^\top\mathrm{vec}(\Sigma^{-1}(\mathbf{s} - \boldsymbol{\mu})\Sigma_T^{-1}) = \mathrm{tr}((\mathbf{s} - \boldsymbol{\mu})^\top\Sigma^{-1}(\mathbf{s} - \boldsymbol{\mu})\Sigma_T^{-1})$$

$$= \mathrm{tr}(\Sigma^{-1}(\mathbf{s} - \boldsymbol{\mu})\Sigma_T^{-1}(\mathbf{s} - \boldsymbol{\mu})^\top).$$

Therefore, the likelihood function becomes:

$$L(\Sigma|\bar{\mathbf{s}}) = (2\pi)^{-d(T-1)/2}|\Sigma_T|^{-d/2}|\Sigma|^{-(T-1)/2}\exp[-\mathrm{tr}(\Sigma^{-1}(\mathbf{s} - \boldsymbol{\mu})\Sigma_T^{-1}(\mathbf{s} - \boldsymbol{\mu})^\top)/2].$$

Taking the logarithm, the log-likelihood function is:

$$\ell(\Sigma|\bar{\mathbf{s}}) = -\frac{d(T-1)}{2}\log(2\pi) - \frac{d}{2}\log(|\Sigma_T|) - \frac{(T-1)}{2}\log(|\Sigma|) - \frac{1}{2}\mathrm{tr}(\Sigma^{-1}(\mathbf{s}-\boldsymbol{\mu})\Sigma_T^{-1}(\mathbf{s}-\boldsymbol{\mu})^\top).$$

For $n$ independent input sequences $\bar{\mathbf{s}}_1, \ldots, \bar{\mathbf{s}}_n$ with lengths $T_1 + 1, \ldots, T_n + 1$, generated by the same $\Sigma$, the total likelihood is:

$$L(\Sigma|\{\mathbf{s}_i\}_{i=1}^n) = \Pi_{i=1}^n L(\Sigma|\mathbf{s}_i).$$

Then the total log-likelihood function is

$$\ell(\Sigma|\{\mathbf{s}_i\}_{i=1}^n) = \sum_{i=1}^n \ell(\Sigma|\mathbf{s}_i)$$

$$= -\frac{d\sum_{i=1}^n(T_i-1)}{2}\log(2\pi) - \frac{d}{2}\sum_{i=1}^n\log(|\Sigma_{T_i}|) - \frac{\sum_{i=1}^n(T_i-1)}{2}\log(|\Sigma|)$$

$$- \frac{1}{2}\sum_{i=1}^n\mathrm{tr}(\Sigma^{-1}(\mathbf{s}_i - \boldsymbol{\mu}_i)\Sigma_{T_i}^{-1}(\mathbf{s}_i - \boldsymbol{\mu}_i)^\top).$$

$\square$

## A.2 Proof of Proposition 2

*Proof.* To find the MLE of $\Sigma$, we need to minimize the negative log-likelihood function, which is equivalent to minimize

$$g(\Sigma) = \sum_{i=1}^n(T_i - 1)\log(|\Sigma|) + \sum_{i=1}^n\mathrm{tr}(\Sigma^{-1}(\mathbf{s}_i - \boldsymbol{\mu}_i)\Sigma_{T_i}^{-1}(\mathbf{s}_i - \boldsymbol{\mu}_i)^\top).$$

Since $\Sigma = \mathbf{W}\mathbf{W}^\top$ is positive definite, we can compute the gradient of $g(\Sigma)$ with respect to $\Sigma$. Note that $(\mathrm{d}/\mathrm{d}\Sigma)\log(|\Sigma|) = (\Sigma^\top)^{-1} = \Sigma^{-1}$ and

$$\frac{\mathrm{d}}{\mathrm{d}\Sigma}\mathrm{tr}(\Sigma^{-1}(\mathbf{s}_i - \boldsymbol{\mu}_i)\Sigma_{T_i}^{-1}(\mathbf{s}_i - \boldsymbol{\mu}_i)^\top)$$

$$= -(\Sigma^{-1}(\mathbf{s}_i - \boldsymbol{\mu}_i)\Sigma_{T_i}^{-1}(\mathbf{s}_i - \boldsymbol{\mu}_i)^\top\Sigma^{-1})^\top = -\Sigma^{-1}(\mathbf{s}_i - \boldsymbol{\mu}_i)\Sigma_{T_i}^{-1}(\mathbf{s}_i - \boldsymbol{\mu}_i)^\top\Sigma^{-1}.$$

We compute the gradient:

$$\frac{\mathrm{d}}{\mathrm{d}\Sigma} g(\Sigma) = \Big( \sum_{i=1}^{n} (T_i - 1) \Big) \Sigma^{-1} - \Sigma^{-1} \Big( \sum_{i=1}^{n} (\mathbf{s}_i - \boldsymbol{\mu}_i) \Sigma_{T_i}^{-1} (\mathbf{s}_i - \boldsymbol{\mu}_i)^{\top} \Big) \Sigma^{-1}.$$

Setting the gradient to zero for minimization, we have:

$$\widehat{\Sigma} = \Big( \sum_{i=1}^{n} (T_i - 1) \Big)^{-1} \Big( \sum_{i=1}^{n} (\mathbf{s}_i - \boldsymbol{\mu}_i) \Sigma_{T_i}^{-1} (\mathbf{s}_i - \boldsymbol{\mu}_i)^{\top} \Big).$$

$\square$

# B  TRAINING DETAILS

## B.1  SP ENCODER

The WikiSection SP Encoder was trained on 1 A100 GPU for about 10 hours using the training set of WikiSection for 100 epochs. We used SGD optimizer and set the learning rate to be *1e-9*. The $\epsilon$ in the loss function $L_{\mathrm{NLL}}$ is chosen as *1e-7*. The WikiText SP Encoder was trained on 4 A100 GPUs for roughly 20 hours for 4 epochs with WikiText dataset. For this dataset, we trained with AdamW optimizer with learning rate *1e-9* and batch size *32*. The $\epsilon$ in the loss function $L_{\mathrm{NLL}}$ is chosen as *1e-3*. Other hyperparameters can be accessed from the configuration file in the submitted code. Our empirical results show incorporating $\hat{\sigma}_j$ into the $\widehat{\Sigma}_j$ makes no significant results in the downstream tasks, thus we disregard $\hat{\sigma}_j$ during encoder training.

## B.2  HYPER-PARAMETER TUNING

While training the WikiSection SP Encoder, we experimented with different $\epsilon$ in $L_{\mathrm{NLL}}$ to see its impact on the performance of the trained encoder. Note that $\epsilon$ determines the perturbation added to the matrix $\widehat{\Sigma}$. The eigenvalues of the initial $\widehat{\Sigma}$ range from $10^{-6}$ to $10^{-1}$, with the majority of which lying in $[10^{-3}, 10^{-5}]$. Thus we tested the following three different $\epsilon$:

- Large $\epsilon = 10^{-3}$ that is larger that most eigenvalues of $\widehat{\Sigma}$.
- Medium $\epsilon = 10^{-5}$ that is about the same scale of most eigenvalues of $\widehat{\Sigma}$.
- Small $\epsilon = 10^{-7}$ that is smaller than most eigenvalues of $\widehat{\Sigma}$.

We choose the small $\epsilon$ based on the performance.

# C   OTHER SCORES USED IN THIS PAPER

**Entity Grid**   Barzilay & Lapata (2005) is the most recognized entity-based approach. It creates a two-way contingency table for each input document to track the appearance of entities in each sentence. We use Stanford's CoreNLP to annotate the documents and the implementation provided in the Coheoka library[2] to obtain the Entity Grid score.

**Unified Coherence**   Moon et al. (2019) presents a neural-based entity-grid method that integrates sentence grammar, inter-sentence coherence relations, and global coherence patterns, achieving state-of-the-art results in artificial tasks.

**BBScore**   Sheng et al. (2024) introduces BBScore, and also check the main text for a comprehensive comparison between BBScore and SPM.

**SPM + CLS**   We use SPM and local moving window SPM with window sizes being 1, 2, 3, 5, and 10 as features showing how coherent the text is and an conceptual figure is illustrated in Figure 3 . A three-layer perceptron ($1 \times 2048 \times 512 \times 1$ for global task and $1 \times 512 \times 1024 \times 1$ for local task) then takes those features as input and outputs predictions for designed tasks.

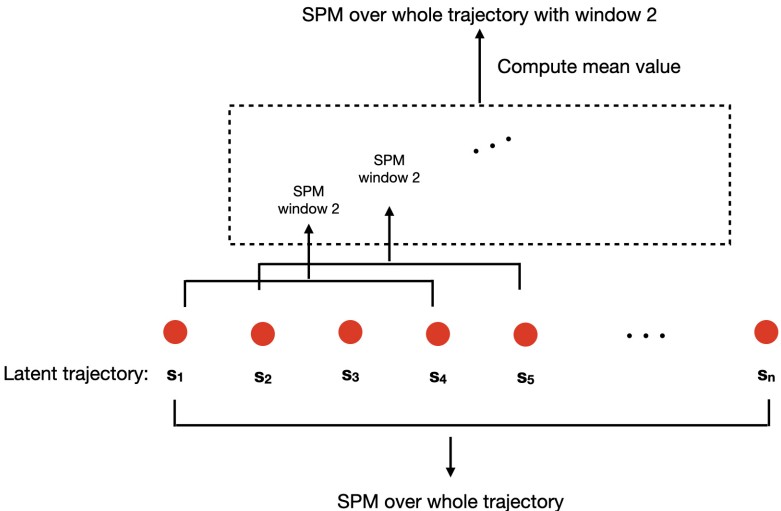

Figure 3: Illustration of SPM over the entire trajectory and SPM with a local moving window (size=2).

---

[2]`https://github.com/kigawas/coheoka`

## D    BBSCORE AND SPM UNDER DIFFERENT ARTICLE LENGTH

As illustrated in Fig 4, we present a visualization of both the BBScore and SPM for the original and block-shuffled scores. From the plot, it is evident that the SPM remains more stable, regardless of the article length. Notably, using SPM, the shuffled scores (in red) exhibit a similar pattern to the unshuffled scores under different article length.

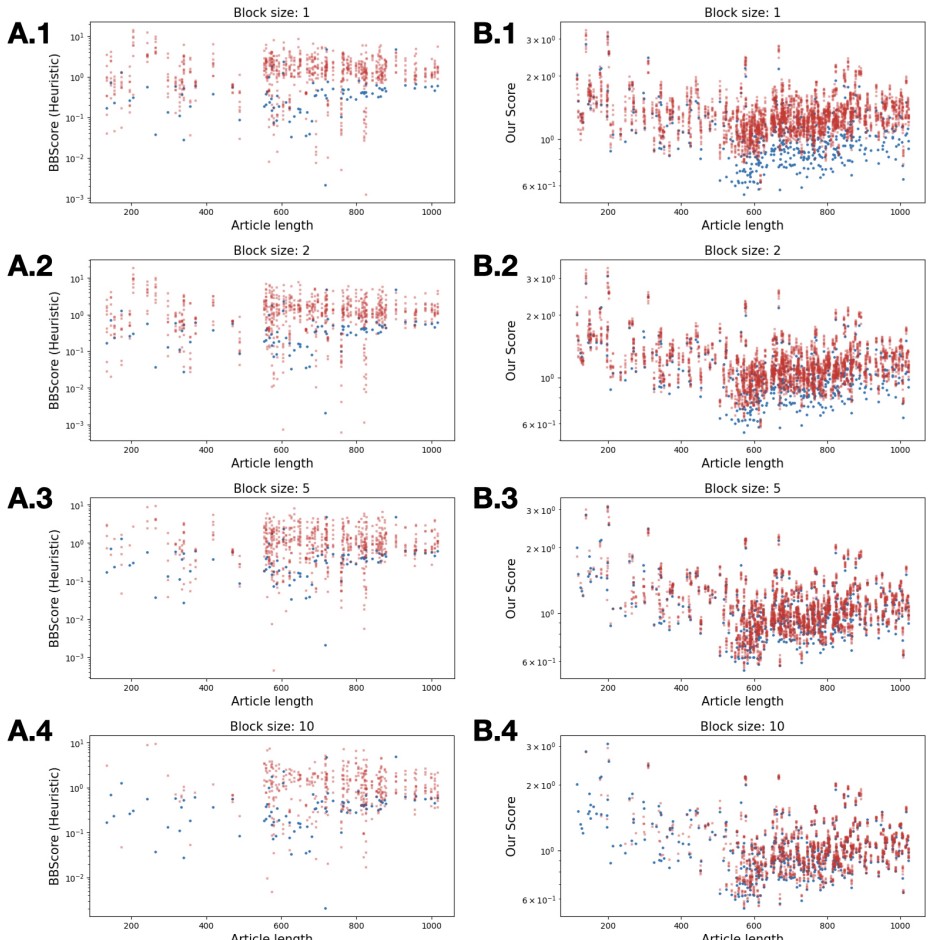

Figure 4: Visualization of BBscore and SPM.

# E    COMPARISON OF LATENTS ENCODED BY CL AND SP ENCODER

First, we conduct a comparison between the BB encoder and the CL encoder. As shown in Fig 5 (A.1) and (B.1), we plot the rough shape of the latent trajectory (blue and red regions) and the target BB (green region), where we observe that the BB encoder appears to learn the target BB more accurately.

To further highlight the presence of ineffective dimensions in the CL Encoder, Figures 5 (A.2) and (B.2) display the latent trajectory of a sample article encoded by the CL Encoder, which reveals only two highly symmetric effective dimensions. In contrast, the SP Encoder captures a more diverse and nuanced trajectory, reflecting superior dimensional effectiveness. This observation may also elucidate why Wang et al. (2022) could not establish a clear relationship between encoder dimensions and the performance of the results.

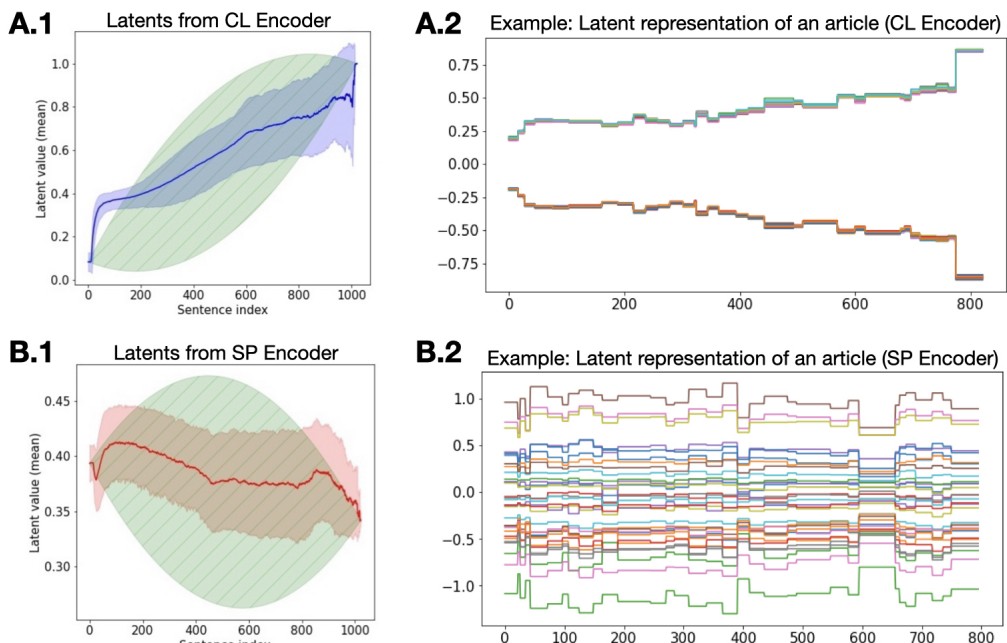

Figure 5: CL Encoder vs SP Encoder in the latent representation: Latent analysis of the CL Encoder (A.1 and A.2) and SP Encoder (B.1 and B.2). (A.2) and (B.2) show an example of the latent representation of an article, with different colors representing different dimensions (total dimension=32).

# F    DIFFERENCE BETWEEN **WIKISECTION** AND **WIKITEXT**

The **WikiSection** dataset comprises 2,165 articles describing cities from Wikipedia, while **WikiText** includes 29,061 featured or high-quality articles covering a broader range of topics. The **WikiSection** dataset is most similar to the "places" category in **WikiText**, which contains approximately 500 articles. To ensure dataset exclusivity, we used string match to check the overlapping. The regular expression query we used is `' (a|the) ([\w\s]*)?(city|town) in'` as it is contained in 1,721 articles out of 2,165 in WikiSection dataset. Using the same query, we examined the WikiText dataset and checked the intersection of first word of the article from both search result. After manually getting rid of false positives, there are around 30 documents found overlap in both datasets. We argue that with that amount of ($\sim$0.1%) contamination, **WikiSection** can be considered out of domain of **WikiText**.

# G HUMAN-AI COMPARISON TEST

Table 6 presents the performance of SPM computed with different $\widehat{\Sigma} \in \mathbb{R}^d$, while Table 7 shows the performance of the BBScore with various $\widehat{\sigma} \in \mathbb{R}$, where the subscript indicates the dataset used for approximation.

The clear improvement over the BBScore demonstrates that accurately capturing structural and temporal information can significantly enhance the model's accuracy. Table 8 display the performance of DetectGPT with more inferences which significantly improves its performance while also takes much longer time to infer.

|  | Human AI comparison | | | Human AI comparison with Q&A | | |
|---|---|---|---|---|---|---|
|  | Human ($\widehat{\Sigma}_{human}$) | Human ($\widehat{\Sigma}_{ai}$) | Human ($\widehat{\Sigma}_{wiki}$) | Human ($\widehat{\Sigma}_{human}$) | Human ($\widehat{\Sigma}_{ai}$) | Human ($\widehat{\Sigma}_{wiki}$) |
| AI ($\widehat{\Sigma}_{human}$) | 70.07 | 70.55 | - | 69.00 | 69.60 | - |
| AI ($\widehat{\Sigma}_{ai}$) | 59.98 | 61.52 | - | 58.19 | 59.74 | - |
| AI ($\widehat{\Sigma}_{wiki}$) | - | - | **70.67** | - | - | **69.71** |

Table 6: Combined accuracy of human AI comparison and human AI comparison with Q&A

|  | Human AI comparison | | | Human AI comparison with Q&A | | |
|---|---|---|---|---|---|---|
|  | Human ($\widehat{\sigma}_{human}$) | Human ($\widehat{\sigma}_{ai}$) | Human ($\widehat{\sigma}_{wiki}$) | Human ($\widehat{\sigma}_{human}$) | Human ($\widehat{\sigma}_{ai}$) | Human ($\widehat{\sigma}_{wiki}$) |
| AI ($\widehat{\sigma}_{human}$) | 35.99 | **45.13** | - | 35.04 | **38.84** | - |
| AI ($\widehat{\sigma}_{ai}$) | 26.37 | 37.05 | - | 33.73 | 38.12 | - |
| AI ($\widehat{\sigma}_{wiki}$) | - | - | 37.53 | - | - | 31.47 |

Table 7: Human-AI Task Results with BBScore (Sheng et al., 2024).

|  | Human AI comparison | | Human AI comparison with Q&A | |
|---|---|---|---|---|
| Number of Perturbations | 1 | 10 | 1 | 10 |
| Number of LLM Inferences | Number of Perturbations + 1 | | | |
| Accuracy | 64.30 | 84.89 | 63.30 | 83.13 |

Table 8: Human-AI Task Results with DetectGPT (Mitchell et al., 2023). As a comparison, SPM only requires one LLM inference.

# H  COMPUTATION EFFICIENCY ANALYSIS OF SPM

For the computational efficiency analysis of SPM, as shown in Figure 6, the y-axis represents computation time, while the x-axis indicates article length. The theoretical computational complexity of SPM is $O(T^2)$, primarily due to matrix multiplications inherent in its definition. This complexity is fundamental to fully leveraging temporal information for sequence evaluation. Empirically, the observed computation time is slightly better than the theoretical prediction, thanks to the computational acceleration provided by Numpy. These results demonstrate that SPM is not only feasible for real-time applications but also retains its robust evaluation capabilities.

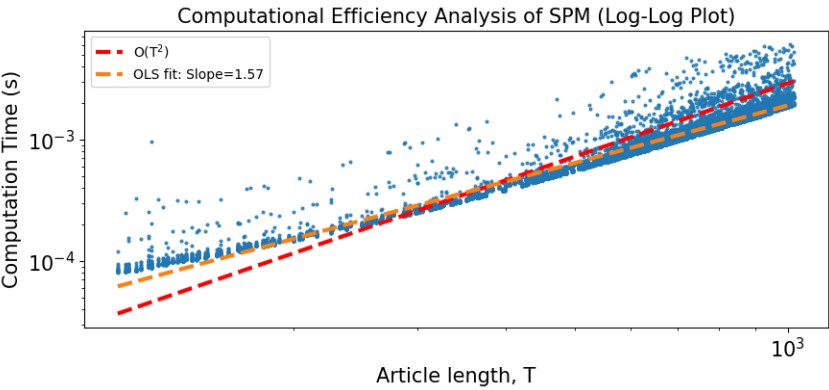

Figure 6: The computation time of SPM for different article lengths. It reveals a quadratic relationship (experimentally 1.57, theoretically 2) between article length and computation time, with each article processed in approximately $\sim 10^{-3}$ seconds.

# I  COMPARISON WITH RECENT LLM

Due to time and computational resource constraints, we tested our model using LLaMA3-1B and LLaMA3-3B, with GPT2-117M which is the LLM model used in the main section. The results are summarized in the table below. Specifically, we compare the following two tasks:

1. **Shuffle Test (Global):** LLaMA3-3B outperforms both GPT2-117M and the state-of-the-art method (Moon et al., 2019), demonstrating its effectiveness in capturing global sequence structure.

2. **Mixed Shuffle Test:** LLaMA3-3B surpasses GPT2-117M for smaller blocks (b=1, b=2), but its performance decreases for larger blocks (b=5, b=10). This may be attributed to our approach, where only an MLP layer was trained without fine-tuning the LLMs. Consequently, GPT2 might better capture certain latent space properties, leading to a more balanced performance across tasks.

For models of the same type, increased parameter sizes consistently yield better performance. However, in Mixed Shuffled Task, examining the performance drop from b=1 to b=10 reveals an interesting pattern: LLaMA3-3B exhibits a sharper decrease (26%) compared to LLaMA3-1B (18%) and GPT2-117M (12%). This suggests a trade-off where larger models excel at capturing local details (b=1) but might sacrifice robustness for global structures (b=10). This insight highlights an intriguing direction for future exploration — different LLM architectures may facilitate learning stochastic representations in task-specific ways.

Although our paper focuses on theoretical foundations and experimental validation, these findings with recent LLMs provide additional evidence supporting our work's ability to decode spatial and temporal information from stochastic representations. They further validate our argument that the fitness of distribution can be leveraged for various downstream tasks and demonstrate the broader potential of our approach in addressing diverse challenges with advanced LLMs.

| Tasks | SHUFFLED TEST (GLOBAL) | | | | MIXED SHUFFLED TEST | | | |
|---|---|---|---|---|---|---|---|---|
| | $\mathcal{D}_{b=1}$ | $\mathcal{D}_{b=2}$ | $\mathcal{D}_{b=5}$ | $\mathcal{D}_{b=10}$ | $\mathcal{D}_{b=1}$ | $\mathcal{D}_{b=2}$ | $\mathcal{D}_{b=5}$ | $\mathcal{D}_{b=10}$ |
| **GPT2-117M** | 95.06 | 94.72 | 95.13 | 95.67 | 90.32 | 86.03 | **79.26** | **77.89** |
| **LLaMA3-1B** | 93.21 | 90.42 | 86.76 | 86.55 | 80.30 | 72.68 | 66.39 | 62.44 |
| **LLaMA3-3B** | **99.57** | **98.75** | **98.14** | **98.74** | **95.04** | **86.46** | 74.00 | 69.06 |

Table 9: Different tasks with SPM