# OpenReview forum: "On the Sequence Evaluation based on Stochastic Processes"
_ICLR.cc/2025/Conference — Submitted to ICLR 2025_

### Official Review · Reviewer_fsYG · 2024-10-27

**Soundness:** 2
**Presentation:** 3
**Contribution:** 2
**Rating:** 3
**Confidence:** 4

**Summary:**

This paper explores the use of Brownian Bridges as a type of language model encoding. Specifically the model is used to represent the temporal dynamics in embedding space. The hope is that non-typical generations that lack structural coherence will be easy to discriminate from generations that have a more typical coherence pattern. To do this the authors fit a BB model on to the output of GPT-2, and then evaluate other documents (synthetic and natural) in an unsupervised manner.

The use of BB models has been previously explored, most notably in "Language modeling via stochastic processes" (Wang, 2022). The authors distinguish this work from the previous work by switching to a non-identity covariance term and by fitting the model using MLE versus a contrastive approximation.

**Strengths:**

* The technical presentation of this work is strong. It provides a clear explanation of the methods used and how they can be adapted to a text evaluation task. The argument for non-diagonal \Sigma and to attempt full MLE is reasonable, and the motivation for using this approach is valid.

* While not a novel method per se, this line of research is underexplored compared to other papers in the language modeling space. It would be good to see more creative uses of these methods in practice.

* The zero-shot performance of human-AI detection seems promising. It is good that a well motivated probabilistic model can discriminate systems based on purely statistical properties (although would be curious to see the results on more recent LLMs)

**Weaknesses:**

* The applications in the work are primarily of moderate-to-low interest as applications in NLP. While the Shuffle test and Entity Grid based models are of critical historical importance in NLP they are not actively used in modern systems. The results given are similar to other approaches for these tasks and serve primarily to validate the approach as opposed to improve metrics.

* The zero shot detection task from detect-GPT is a bit more promising, but it was unclear to me why this method was only applied on a small subset of the data sets that were used in that paper.

* Results comparing the main contribution of the work MLE based SP seem to be mixed-to-negative? It would be useful to have a better sense of how the authors see these results and if they justify the more costly procedure.

* The main thing lacking from the work seems to be any mention of text generation. Given that Wang 2022 is able to generate it is unclear to me why these results are not included or studied.

* It seems as if there is a contrastive style triplet approximation being used during MLE training. Given that step, it is less clear to me what the advantages are as an algorithm with contrastive loss.

**Questions:**

(just the ones above)

---

> ### Author Response · Authors · 2024-11-19
> **Response to Weakness 1,2,3**
>
> **Response to Weakness 1:**  We fully acknowledge the limitations of existing coherence evaluation methods and agree that they must be applied with caution. Consequently, in our study, we employed these methods to validate our primary contribution — demonstrating that the temporal and structural dependencies, as learned from stochastic representations, can generalize across a wide range of tasks, including coherence evaluation. As far as we know, there has not been a coherence evaluation metric designed based on the likelihood of stochastic representations. Therefore, a fundamental step was to validate our proposed metric using well-accepted and robust methods currently available.
>
> As you pointed out, there are inherent limitations to these standard methods. Through our own analysis, we identified a significant issue: the challenge of comparing coherence across articles of different lengths. Article length has been a persistent issue in coherence evaluation, and it also presents a core challenge in text generation. To address this, we extended the Shuffle Test by developing a Mixed Shuffle Test aimed at assessing whether our metric is independent of article length. As shown in Table 2, our score (SPM) successfully evaluates coherence between articles of varying lengths — a task that even well-established methods like the Entity Grid models struggle with.
>
> Furthermore, the visualization of score distributions for both SPM and BBScore in Figure 4 reinforces our contribution. By better approximating the dependencies between latent variables and implementing a novel design, our score (SPM) demonstrates consistent robustness across various article lengths. This consistency provides strong evidence for its insensitivity to length variations, underscoring the reliability of our metric and its potential for broader applicability in diverse tasks. Given the demonstrated efficacy validated in this study and the multifaceted improvements over existing baselines, the integration of SPM into contemporary large language model (LLM) training paradigms holds significant promise. For instance, as shown in Table 9 (Appendix), due to time and computational constraints, we only present additional results with LLaMA3-1B and 3B, where we replace GPT2-117M with LLaMA3. These results reveal a clear improvement, even surpassing the state-of-the-art method (Moon et al., 2019) discussed in the manuscript. Although our key focus of this manuscript is on theoretical foundations and experimental validation, these findings with recent LLMs further substantiate our work’s ability to decode spatial and temporal information from stochastic representations. They reinforce our argument that distribution fitness can be leveraged for various downstream tasks and highlight the broader potential of our approach in addressing challenges with advanced LLMs. Apart from this, incorporating SPM within model alignment frameworks, such as Reinforcement Learning from Human Feedback (RLHF), could potentially enhance model preference alignment. This approach would regulate long-form model outputs by guiding generation trajectories to adhere to a Brownian Bridge structure within the latent space.
>
> **Response to Weakness 2:** In fact, as discussed in Section 3.2, we clearly stated that our encoders were trained on the WikiSection dataset (distinct from HC3) and then tested on HC3 for human-AI discrimination. However, this does not imply that our evaluation was limited to a small subset of the HC3 dataset.
>
> **Response to Weakness 3:** We appreciate the reviewer’s concern. Empirically, the SPM consistently outperforms BBScore. Regarding the SP Encoder compared to the CL Encoder, we acknowledge that the results appear mixed. However, the training time for the two encoders is comparable, as both involve sampling a triplet of time points. In terms of performance, when focusing on O.O.D. tasks, which are inherently more challenging, the SP Encoder demonstrates clear advantages over the CL Encoder. This is because the SP Encoder effectively captures both the distribution and the dynamics of the entire article. This capability enables SPM to robustly evaluate and identify O.O.D. stochastic representations. Additionally, we encourage you to refer to our response to comment **Weakness 5**, where we further highlight the advantages of the SP Encoder, supported by both theoretical justifications and empirical evidence.

---

> ### Author Response · Authors · 2024-11-19
> **Response to Weakness 4,5**
>
> **Response to Weakness 4:**  We acknowledge that text generation is a critical task in NLP and a central topic in generative AI. However, this paper focuses on developing a theoretical framework for modeling long sequences through stochastic representations and introduces the SPM score to uncover insights derived from distributional "matching" within this representation for various downstream tasks. While we demonstrate two downstream applications—coherence evaluation and human-AI differentiation—these are illustrative examples rather than the central theme of the paper. Specifically, we did not address text generation for the following reasons:
>
> 1. **Focus on Theoretical Contributions**:  The paper’s primary contributions lie in its novel approach to modeling temporal and structural correlations within sequences via stochastic representations and the introduction of a new score function. These contributions focus on the encoder aspect of neural networks. Including discussions on text generation would shift attention toward the decoding process, which falls outside the scope of the paper.
>
> 2. **Challenges in Text Generation Evaluation**:  Evaluating the quality of open-domain text generation remains inherently challenging with automatic metrics, while rigorous human evaluations require significant resources. Addressing this topic adequately would dilute the paper’s core contributions.
>
> Notably, our framework bears conceptual similarities to diffusion models in dynamic flow learning [1]. However, our work addresses dynamic flow within the latent space of language models rather than explicit data domains, making the problem fundamentally more complex and intractable. We aim to open a new direction by extending the concept of distribution evaluation, widely explored in diffusion models, to broader applications within language models.
>
> As highlighted in our response to **Weakness 1**, future work will investigate leveraging SPM as a guidance score to train reward models and assess the coherence of LLM-generated outputs. This will involve aligning outputs with stochastic processes (e.g., Brownian bridges) in the latent space, thereby extending the utility of our framework to text generation tasks.
>
> **Response to Weakness 5:** Although the training of the SP Encoder also involves sampling a triplet of time points, its primary goal is fundamentally different from that of the CL Encoder. In the SP Encoder, we use the negative log-likelihood as the loss function, ensuring that the encoded sequence conforms to the desired stochastic representation. The triplet sampling procedure is introduced to accelerate the training process. Furthermore, likelihood-based training is more natural in this context since our objective is to ensure the encoded sequence follows the specified distribution, rather than performing prediction tasks. The advantages of our SP Encoder are also evident both empirically. In Figure 2, panels (A) and (B) show that the covariance matrix for the CL Encoder exhibits high similarity across dimensions, indicating the presence of only a few effective dimensions. In contrast, the SP Encoder significantly reduces this similarity, suggesting a more effective utilization of all dimensions. Panel (C) highlights the robustness of the representations generated by the SP Encoder. Figure 5 further illustrates the latent trajectory of a sample article. While the CL Encoder reveals only one effective dimension due to symmetry across two dimensions, the SP Encoder captures a more diverse and nuanced trajectory, demonstrating superior dimensional effectiveness. Therefore, the SP Encoder provides a more robust and effective encoding framework compared to CL Encoder.
>
> ---
>
> **Reference**:
> [1]: Chidambaram, M., Gatmiry, K., Chen, S., Lee, H., & Lu, J. (2024). *What does guidance do? A fine-grained analysis in a simple setting*. arXiv preprint arXiv:2409.13074.

---

> ### Author Response · Authors · 2024-11-25
> **Look forward to your response**
>
> Dear Reviewer fsYG,
>
> We hope you have had the opportunity to review our responses and clarifications. As the discussion period is drawing to a close, we would be grateful if you could confirm whether our updates have fully addressed your concerns. Should you have any further comments or questions, we would be more than happy to address them at your convenience.
>
> Thank you once again for your valuable time and thoughtful feedback. We genuinely appreciate your efforts in reviewing our work.
>
> Best regards,
>
> The Authors

---

### Official Review · Reviewer_SYhi · 2024-10-29

**Soundness:** 3
**Presentation:** 3
**Contribution:** 3
**Rating:** 8
**Confidence:** 2

**Summary:**

This paper introduces a new method for evaluating the coherence of long text sequences using a stochastic process model called the Brownian Bridge. The main metric, the Stochastic Process Metric (SPM), captures both time-related and structural details, helping to measure text coherence and distinguish between human and AI-generated text. The paper also introduces an SP Encoder that uses a negative log-likelihood objective, which performs well on out-of-domain tasks and improves text sequence analysis. The results suggest that SPM and the SP Encoder could be valuable for text coherence evaluation and human-AI discrimination tasks.

**Strengths:**

- The paper introduces a novel SPM that effectively captures temporal and structural dependencies, showcasing a new direction in long-sequence evaluation.
- Leveraging the Brownian Bridge model adds theoretical robustness, enhancing coherence evaluation.
Robust Encoder Design: The SP Encoder, with its negative log-likelihood loss, is shown to outperform contrastive learning approaches in out-of-domain tasks, adding versatility.
- The metric’s success in distinguishing human from AI-generated text is a practical application with high relevance.
- SPM is demonstrated to work well across tasks, including mixed shuffle tests, with flexibility across varying text lengths.

**Weaknesses:**

Overall, this paper is good, with no significant weaknesses. It may be further improved by addressing the following considerations:

- The reliance on stochastic processes and likelihood estimation might increase computational demands, which is not thoroughly addressed.
- SPM’s struggles with capturing local coherence perturbations suggest that it may overlook smaller, context-specific coherence challenges.
- The SP Encoder’s success partly relies on domain-specific parameters, which might limit its application across highly diverse domains without retraining.

**Questions:**

- How does SPM perform in real-time applications, given the computational demands of likelihood-based methods?
- How does SPM compare to transformer-based coherence evaluation metrics in terms of both performance and computational efficiency?

---

> ### Author Response · Authors · 2024-11-19
> **Response to Weakness 1,2,3**
>
> **Response to Weakness 1:**  Thank you for raising this point. The additional computational complexity in both the SP Encoder and SPM arises from inverting the structural and temporal matrices. However, these operations are computationally efficient due to their fixed dimensions.
>
> 1. **Structural Matrix:**  In the SP Encoder, the structural matrix, $\widehat{\Sigma}_j^{-1}$, and in SPM, $\widehat{\Sigma}^{-1}$, are both of fixed dimension $d \times d$. Consequently, the computational cost of inversion is negligible.
>
> 2. **Temporal Matrix:**  In the SP Encoder, the temporal matrix, $[\widehat{\Sigma}_{T_i}]_t$, has a fixed dimension of $3 \times 3$ since we randomly sample a triplet of time points. Thus, the cost of inversion during training is negligible. In SPM, the temporal matrix, $\Sigma_T$, does introduce an additional computational cost to accurately capture temporal information when evaluating sequences. This complexity is inherent to our method and not present in previous approaches, which do not account for temporal information. To address this, we propose precomputing and storing the inverse of $\Sigma_T$ for various values of $T$. This optimization requires the inversion to be computed only once, significantly reducing computational overhead when applying SPM to new sequence evaluation tasks.
>
> By incorporating these strategies, we ensure that the proposed SPM remains computationally efficient without compromising its ability to capture both structural and temporal information effectively.
>
> **Response to Weakness 2:** Your comments is correct that for long text, global and local coherence can not always be mutually guaranteed. SPM mainly focuses on global coherence by modeling the whole sequence as a Brownian Bridge process, the addition of $\Sigma_T$ considers temporal dependencies that allows the model to capture the coherence between any two time points along the sequence. We can observe improvements by using this covariance matrix on the local coherence evaluation task comparing to the method in BBScore paper, where they only consider isotropic covariance matrix that fails to temporal correlations. On the other hand, our method is fully unsupervised and requires no reference text, thus can be used to compare text of any length after training the encoder. In the meanwhile, the SOTA model’s requirement for end-to-end training with equal-length paired texts (coherent vs. incoherent) limits its application.
>
> **Response to Weakness 3:** You have rightly highlighted a potential limitation of the SP Encoder and SP Metric, specifically their partial dependence on domain-specific parameters. However, as evidenced by our results in the Human-AI task (Table 3) and the OOD task (Table 5), our design achieves comparable or even superior performance when compared to both baseline and SOTA methods. This strong performance is attributed to the fact that our SP design not only effectively captures domain-specific features but also learns a robust stochastic representation — what we refer to as the temporal and structural properties of text. These properties are intrinsic and broadly shared across various long texts, enabling our SP design to generalize well, even to previously unseen datasets.

---

> ### Author Response · Authors · 2024-11-19
> **Response to Question 1,2**
>
> **Response to Question 1:** To address this concern, we precompute and store the inverse of $\Sigma_T$ for various values of $T$, as discussed in our response to the **Weakness 1**. This optimization significantly reduces the computational overhead during real-time applications. We also provide a computational efficiency analysis of SPM in Figure 6 (Updated pdf), where the y-axis represents computation time, and the x-axis represents article length. The theoretical computational complexity of SPM is $O(T^2)$, primarily due to matrix multiplications in its definition. This complexity is unavoidable if we aim to fully utilize the temporal information for sequence evaluation. Empirically, the observed computation time is slightly better than the theoretical complexity. This improvement stems from computation acceleration provided by Numpy, making SPM more efficient in practice. These results demonstrate that SPM is feasible for real-time applications while maintaining its robust evaluation capabilities.
>
> **Response to Question 2:** Thank you for pointing out the need to compare SPM with transformer-based coherence evaluation metrics. SPM is a transformer-based evaluation metric in the sense that it generates text embeddings from GPT. SPM then uses likelihood function to calculate scores based on those embeddings. To our knowledge, the other transformer-based model is proposed by Jeon \& Strube (ACL 2022) to evaluate local coherence. That model uses XLnet to generate embeddings and a custom transformer-based structure to calculate scores. However, it did not outperform the Unified Coherence model (Moon et al., 2019) in the shuffle test. Given this limitation, we believe it is more appropriate and informative to compare SPM with the state-of-the-art Unified Coherence model as the baseline.
>
> In terms of computational efficiency, SPM is designed to be lightweight. Once embeddings are precomputed, the primary computational expense arises from the feedforward operations of the MLP. In contrast, both the transformer-based model and the LSTM-based Unified Coherence model require more computationally intensive operations to process hierarchical or sequential structures, such as multi-head attention and recurrent computations. In conclusion, SPM achieves strong performance with significantly higher computational efficiency than transformer-based or LSTM-based models.

---

> ### Author Response · Authors · 2024-11-25
> **Follow-Up on Discussion and Clarifications**
>
> Dear Reviewer SYhi,
>
> Thank you very much for your positive feedback on our work! We hope you have had the chance to review our responses and clarifications. As the discussion period is drawing to a close, we would greatly appreciate it if you could confirm whether our updates have fully addressed your concerns.
>
> Thank you again for your time and thoughtful review.
>
> Best regards,
>
> The Authors

---

> > ### Comment · Reviewer_SYhi · 2024-11-26
> >
> > Thank you for the authors’ response, which has addressed my concerns. I still think that this paper is of high quality and presents an interesting idea. I will maintain my score, and I wish you the best of luck.

---

### Official Review · Reviewer_NuAY · 2024-11-05

**Soundness:** 2
**Presentation:** 1
**Contribution:** 2
**Rating:** 3
**Confidence:** 4

**Summary:**

This paper explores the structural and temporal characteristics encoded in the stochastic representation of latent trajectories and their applications in NLP tasks through theoretical and empirical studies. It introduces a flexible coherence evaluation metric (SPM) that is not influenced by individual article properties like text length. To verify this, the authors designed a Mixed Shuffle test based on the established Shuffle test. They also discovered that their metric, which evaluates the fit of target stochastic processes, can help distinguish between human-written and AI-generated data, indicating that the stochastic representation encodes useful properties for human-AI discrimination.

**Strengths:**

1. The SP Encoder's performance on out-of-domain datasets highlights its robustness and generalizability.
2. The theoretical analysis is solid.

**Weaknesses:**

1. The paper does not adequately address the relationship with BBScore, another metric that uses a Brownian Bridge-Based approach for sequence evaluation. This omission limits the perceived novelty of the proposed method.
2. The introduction and methodology sections are not well organized, making it difficult to follow the flow of ideas and understand the proposed approach.
3. The experiments only use GPT-2 as the backbone model, which limits the generalizability of the findings. To provide a more comprehensive evaluation, the proposed methods should be tested on a wider range of representative large language models (LLMs), such as LLaMA and others.

**Questions:**

See above

---

> ### Author Response · Authors · 2024-11-19
> **Response to Weaknesses and Questions**
>
> **Response to Weakness 1:** We appreciate the reviewer’s concern regarding the relationship between BBScore and our proposed SPM. Compared to BBScore, SPM is the **first** method to utilize both **temporal** and **structural** information for evaluating encoded sequences. In contrast, BBScore does not incorporate temporal information, and the encoder it uses (CL Encoder) does not account for structural information. SPM introduces a distinct stochastic representation and encoder, resulting in a fundamentally different definition and implementation from BBScore. Specifically, the SP Encoder employed in SPM encodes both structural and temporal aspects of sequences, which are integral to the definition of SPM. These enhancements are rigorously justified from a theoretical perspective in Section 2. Additionally, the superiority of SPM over BBScore is demonstrated empirically in several experiments outlined in Section 4. Given these theoretical and empirical distinctions, we believe the novelty of SPM and its advantages over BBScore are clearly established in the paper.
>
> **Response to Weakness 2:** We acknowledge the importance of clear organization to facilitate readability and understanding. In the introduction, we first discuss the use of stochastic representations in generative models and the application of the Brownian Bridge in stochastic representation modeling. To ensure a smoother reading experience, related work discussions are all moved to Section 5, avoiding interruptions to the primary narrative. The introduction then transitions to presenting our two main contributions: the SPM and the SP Encoder. We conclude this section with a concise summary of these contributions. In the methodology section, we explain the stochastic representation, focusing on how Brownian Bridges are used to model sequences—a foundational concept for our contributions. This is followed by detailed discussions of the SP Encoder and SPM, aligning with the schematic overview provided in Figure 1. The order of these components mirrors the logical flow of Figure 1, designed to help readers grasp the concepts effectively.
>
> **Response to Weakness 3:** The reviewer is correct that it would be more comprehensive to include additional backbone models for evaluation. However, we argue that this work primarily focuses on the theoretical development of a novel model to capture temporal representations for long sequences using a transformer-based architecture.  As GPT-2 is one of the most lightweight decoder-only transformers, it serves as a proof of concept that this theoretical framework aligns with the empirical results. The results in the paper suggest that the proposed SPM framework effectively captures the dynamics of long sequences and demonstrates success in both coherence evaluation and human-AI discrimination.  Recent work has demonstrated scaling laws in large language models (LLMs) as text embedders [1,2,3]. Our work differs from these studies by focusing on learning text dynamics, rather than semantics, in the hidden space.   Upon validation of our theoretical foundations, targeting metric improvements by testing LLMs with varying parameter sizes presents a promising future research direction.
>
> ---
>
> **Reference**:
> [1]: Zhang, X., Li, Z., Zhang, Y., Long, D., Xie, P., Zhang, M., & Zhang, M. (2023). *Language models are universal embedders*. arXiv. [https://arxiv.org/abs/2310.08232](https://arxiv.org/abs/2310.08232).
> [2]: Muennighoff, N. (2022). *SGPT: GPT sentence embeddings for semantic search*. arXiv. [https://arxiv.org/abs/2202.08904](https://arxiv.org/abs/2202.08904).
> [3]: Ni, J., Qu, C., Lu, J., Dai, Z., Hernandez Abrego, G., Ma, J., Zhao, V., Luan, Y., Hall, K., Chang, M.-W., & Yang, Y. (2022). *Large dual encoders are generalizable retrievers*. In Y. Goldberg, Z. Kozareva, & Y. Zhang (Eds.), Proceedings of the 2022 Conference on Empirical Methods in Natural Language Processing (pp. 9844–9855). Association for Computational Linguistics. [https://doi.org/10.18653/v1/2022.emnlp-main.669](https://doi.org/10.18653/v1/2022.emnlp-main.669).

---

> ### Author Response · Authors · 2024-11-21
> **Follow-up on Weakness 3: LLaMA Result**
>
> Due to time and computational resource constraints, we tested our framework using **LLaMA3-1B** and **LLaMA3-3B**, and compare these results with **GPT2-117M** which is the LLM model used in the manuscript. The results are summarized in the table below. Specifically, we compare the following two tasks:
>
> 1. **Shuffle Test (Global):** LLaMA3-3B outperforms both GPT2-117M and the state-of-the-art method (Moon et al., 2019) used in our manuscript, demonstrating its effectiveness in capturing global sequence structure.
>
> ### Shuffled Test (Global)
>
> | **Tasks**      | $\mathcal{D}_{b=1}$ | $\mathcal{D}_{b=2}$ | $\mathcal{D}_{b=5}$ | $\mathcal{D}_{b=10}$ |
> |----------------|---------------------|---------------------|---------------------|----------------------|
> | **GPT2-117M**  | 95.06               | 94.72               | 95.13               | 95.67                |
> | **LLaMA3-1B**  | 93.21               | 90.42               | 86.76               | 86.55                |
> | **LLaMA3-3B**  | **99.57**           | **98.75**           | **98.14**           | **98.74**            |
>
> 2. **Mixed Shuffle Test:** LLaMA3-3B surpasses GPT2-117M for smaller blocks (`b=1`, `b=2`), but its performance decreases for larger blocks (`b=5`, `b=10`). This may be attributed to our approach, where only an MLP layer was trained without fine-tuning the LLMs. Consequently, GPT2 might better capture certain latent space properties, leading to a more balanced performance across tasks.
>
> ### Mixed Shuffled Test
>
> | **Tasks**      | $\mathcal{D}_{b=1}$ | $\mathcal{D}_{b=2}$ | $\mathcal{D}_{b=5}$ | $\mathcal{D}_{b=10}$ |
> |----------------|---------------------|---------------------|---------------------|----------------------|
> | **GPT2-117M**  | 90.32               | 86.03               | **79.26**           | **77.89**            |
> | **LLaMA3-1B**  | 80.30               | 72.68               | 66.39               | 62.44                |
> | **LLaMA3-3B**  | **95.04**           | **86.46**           | 74.00               | 69.06                |
>
> For models of the same type (LLaMA3), increased parameter sizes consistently yield better performance. However, in the Mixed Shuffled Task, examining the performance drop from `b=1` to `b=10` reveals an interesting pattern: LLaMA3-3B exhibits a sharper decrease (26%) compared to LLaMA3-1B (18%) and GPT2-117M (12%). This suggests a trade-off where larger models excel at capturing local details (`b=1`) but might sacrifice robustness for global structures (`b=10`). This insight highlights an intriguing direction for future exploration — different LLM architectures may facilitate learning stochastic representations in task-specific ways.
>
> Although our paper focuses on theoretical foundations and experimental validation, these findings with recent LLMs provide additional evidence supporting our work's ability to decode spatial and temporal information from stochastic representations. They further validate our argument that the fitness of distribution can be leveraged for various downstream tasks and demonstrate the broader potential of our approach in addressing diverse challenges with advanced LLMs.

---

> ### Author Response · Authors · 2024-11-25
> **Look forward to your response**
>
> Dear Reviewer NuAY,
>
> We hope you have had the opportunity to review our responses and clarifications. As the discussion period is nearing its conclusion, we would greatly appreciate it if you could confirm whether our updates have adequately addressed your concerns.
>
> Thank you for your time and consideration.
>
> Best regards,
>
> The Authors

---

> > ### Comment · Reviewer_NuAY · 2024-11-26
> >
> > Thank you for your response, but I would like to keep my score.

---

### Author Response · Authors · 2024-11-19
**Global Response**

We sincerely appreciate the time and effort each reviewer has dedicated to evaluating our manuscript.

1. Reviewer **NuAY** acknowledged the "robustness and generalizability" of the SPM Encoder and the "solid theoretical analysis."
2. Reviewer **SYhi** praised our work, describing the SP Metric and SP Encoder as "novel," "robust," and "highly relevant," noting that it "opens a new direction in long-sequence evaluation" without any significant weaknesses.
3. Reviewer **fsYG** commended our strong technical presentation and described our application as "promising."

Several important points raised by the reviewers have provided valuable insights that greatly enhanced our manuscript. We have carefully addressed these and made the following key modifications:

1. Conducted a theoretical analysis of computational efficiency, supported by numerical evidence (Reviewer **SYhi**).
2. Added a general comparison with current transformer-based evaluation metrics (Reviewer **SYhi**).
3. Clarified the significance and practical utility of the spatio-temporal structures captured by the stochastic representation (Reviewer **SYhi**, **fsYG**).
4. Differentiated the negative log-likelihood-based SP Encoder from the contrastive learning-based CL Encoder and their benefits (Reviewer **fsYG**).
5. Highlighted the differences between SPM and another Brownian bridge-based approach, BBScore (Reviewer **NuAY**).
6. Explored open directions and potential applications of our theoretical foundation to other LLMs (Reviewer **NuAY**).
7. Added results and analysis with LLaMA3-1B and 3B, providing additional evidence supporting our theoretical framework and arguments (Reviewer **NuAY**).

In summary, our main theoretical work underscores the importance of spatial and temporal information in stochastic representations. The article-length-insensitive SPM and robust SP Encoder proposed in this study open promising avenues for long-text modeling, coherence evaluation, and generation in NLP, including human-AI differentiation and enhancing long-text applications.

We sincerely thank the reviewers for their invaluable feedback and constructive suggestions.

---

### Meta-Review · Area_Chair_dfCx · 2024-12-21

**Metareview:**

This paper proposes a method for evaluating the coherence of text using Brownian Bridge, by first encoding text into latent embeddings, and then fit a Brownian Bridge on the embedding traces. This paper also proposes a training method for learning latent representations. This work demonstrates that the proposed method can be used to distinguish human-written from machine-generated text.

Strengths:
1. Experiment results on detecting machine-generated text are strong.
2. Using stochastic processes for text evaluation is an underexplored research area and there could be potential followup works from this.
3. This method works well on comparing texts of different lengths.

Weaknesses:
1. As pointed out by reviewers, the applications considered in this paper (such as the Shuffle test and Entity Grid based models) might not be of broad interest, except for the machine-generated text detection experiment.

Overall, a primary concern is that the applications in this paper are not be of broad interest to the community other than the Human-AI discrimination task, and I'm recommending reject for the current version. I'd recommend authors to add more practical applications of this approach in future revisions.

**Additional Comments On Reviewer Discussion:**

One reviewer pointed out that the this work only tested GPT2, the authors emphasized that the focus is a theoretical development of a novel method. Another pointed out issues with evaluation settings, and the authors have clarified that evaluation is done on the full test set.

---

### Decision · Program_Chairs · 2025-01-22

Reject